# STONE: A Submodular Optimization Framework for Active 3D Object Detection

**Ruiyu Mao     Sarthak Kumar Maharana     Rishabh K Iyer     Yunhui Guo**
Department of Computer Science
The University of Texas at Dallas, Richardson, TX, USA
{rxm210041, skm200005, rishabh.iyer, yunhui.guo}@utdallas.edu

## Abstract

3D object detection is fundamentally important for various emerging applications, including autonomous driving and robotics. A key requirement for training an accurate 3D object detector is the availability of a large amount of LiDAR-based point cloud data. Unfortunately, labeling point cloud data is extremely challenging, as accurate 3D bounding boxes and semantic labels are required for each potential object. This paper proposes a unified active 3D object detection framework, for greatly reducing the labeling cost of training 3D object detectors. Our framework is based on a novel formulation of submodular optimization, specifically tailored to the problem of active 3D object detection. In particular, we address two fundamental challenges associated with active 3D object detection: data imbalance and the need to cover the distribution of the data, including LiDAR-based point cloud data of varying difficulty levels. Extensive experiments demonstrate that our method achieves state-of-the-art performance with high computational efficiency compared to existing active learning methods. The code is available at https://github.com/RuiyuM/STONE

## 1 Introduction

In many emerging applications such as autonomous driving, it is critical to localize objects in a 3D scene for accurate scene understanding [8, 56]. This is usually achieved by using a 3D detection model based on LiDAR point cloud data with oriented bounding boxes and semantic labels. While highly accurate recognition and localization of objects can be achieved due to recent advancements in deep learning, this performance often comes at the expense of requiring a large volume of labeled point cloud data, which is much more costly to collect compared to typical RGB images [51, 52].

Active learning (AL) is the standard method for reducing labeling costs in machine learning [7, 45]. AL often starts with a small labeled set and iteratively selects the most informative samples for label acquisition from a large pool of unlabeled data, given a labeling budget. The informativeness, of a selected sample, can be measured in various ways. For example, by computing the sampling uncertainty as measured by maximum Shannon entropy [48, 32] or estimated model changes or finding the most *representative* samples to avoid sample redundancy [36, 17, 40] by using greedy coreset algorithms [44, 18] or clustering-based approaches [38, 57, 3, 42].

Active learning has proven highly effective in reducing labeling costs for recognition tasks, but its application in LiDAR-based object detection remains limited and under-explored [11, 22, 43]. Compared to standard recognition tasks, there are two fundamental challenges: **1)** Depending on the specific scene, each category involves different difficulty levels (EASY, MODERATE, or HARD), which are determined by the size, occlusion level, and truncation of 3D objects. *Ideally, the selected labeled point cloud data should include varying difficulty levels.* **2)** Each 3D scene can contain multiple objects, leading to highly imbalanced label distributions in the point cloud data. For example, most

point clouds include cars, but not cyclists. Addressing data imbalance is thus crucial for selecting informative point clouds to train the 3D object detector.

In a recent study, CRB [35] proposed three stages to label unlabeled point clouds hierarchically, ensuring they are concise, representative, and geometrically balanced. One of the critical components of CRB is to achieve label balance in individual point clouds. However, it cannot guarantee the label distribution is balanced across all the labeled point clouds. More recently, KECOR [34] was proposed for characterizing sample informativeness in both classification and regression tasks with a unified measurement. The main idea is to select a subset of point clouds that maximizes the kernel coding rate. This approach enables the selection of representative samples from the unlabeled point cloud. However, it does not address the issue of data imbalance.

In this paper, we propose a **unified** submodular optimization framework, called **STONE**, for active 3D object detection, to address limitations in existing methods. Our framework leverages submodular functions [12], a classical tool for measuring set quality, due to its well-known property of diminishing returns. Building on these fundamental results, we introduce a novel formulation of submodular optimization for active 3D object detection. This formulation not only ensures that the selected point cloud achieves maximal coverage of the unlabeled point cloud but also addresses the issue of data imbalance. Based on the formulation, we propose a two-stage algorithm. Firstly, we select representative point clouds using a submodular function based on gradients computed from hypothetical labels using Monte Carlo dropout [13]. Secondly, we employ a greedy search algorithm to select unlabeled point clouds, aiming to balance the data distribution as measured by another submodular function. Our work makes the following core **contributions**. **1)** We introduce the first submodular optimization framework for active 3D object detection, addressing two fundamental challenges in this domain. **2)** We develop a simple and efficient two-stage algorithm within the framework for selecting representative point clouds and addressing the issue of data imbalance. **3)** We extensively validate the proposed framework on real-world autonomous driving datasets, including KITTI [15] and Waymo Open dataset [52], achieving state-of-the-art performance in active 3D object detection. Furthermore, additional results in active 2D object detection demonstrate the high generalizability of our proposed method.

## 2 Related Works

**Active Learning** (AL) has been a deeply studied topic in machine learning [7, 45], which involves alleviating labeling annotation costs by selecting the most *representative* samples from a pool of unlabeled data, all the while not comprising model performance. Broadly speaking, AL can be categorized into two strategies - *uncertainty* sampling and *representative/diversity* sampling. Algorithms under *representative/diversity* sampling select subsets of data that act as stand-ins or surrogates for the entire dataset [1]. Prior works have explored coreset-based subset selection [44, 18], clustering algorithms [38, 57, 3], and generative adversarial learning [16]. In uncertainty sampling [31], samples are selected by maximizing the Shannon entropy [32] of the posterior probability, minimizing the model's subsequent training error through variance reduction [6], based on the maximum gradient magnitude [46], and considering the disagreement among a committee of model hypotheses [47, 54]. Hybrid sampling strategies are proposed recently [25, 1, 26, 20], combining *representative/diversity* and *uncertainty* sampling, like BADGE [1], where the authors propose to use the gradient magnitude with respect to the classifier's parameters as a measure of uncertainty. This approach selects samples whose gradients cover a wide range of directions.

**Active Learning for 3D Object Detection.** While AL has been actively studied and applied to image classification and regression tasks, it has recently been garnering interest in the 3D object detection community. Initial works like MC-MI [11] make use of Monte-Carlo (MC)-dropout [14] and Deep Ensembles [30] to compute the Shannon entropy in the predicted labels and mutual information between class predictions and model parameters. CONSENSUS [43] estimates uncertainty using ensembles for 2D/3D object detection. While these works relied on generic metrics to measure prediction uncertainty, two other works on 3D object detection, CRB [35] and KECOR [34], were proposed recently. In CRB, the method greedily searches and samples unlabeled point clouds that exhibit concise labels, representative features, and geometric balance. KECOR shows that, for sample selection, maximizing the kernel coding rate is beneficial and improves over task-specific AL methods for 3D detection, at fast-running times.

**Submodular Optimization.** Submodular functions (see Section 3), and thereby optimization, have found wide applications in the selection of data subsets [58, 29], active learning [27, 28], speech recognition [59, 60], continual learning [53], and hyper-parameter tuning [24]. The effectiveness of submodular functions primarily stems from their ability to model diversity via clustering in representation learning. This enhances their capacity to discriminate between different data subsets or classes while ensuring the preservation of unique and relevant features.

## 3 Background

**3D Object Detection.** LiDAR-based 3D object detection aims to localize and recognize objects in point cloud data using oriented bounding boxes and semantic labels. Point clouds are typically generated by LiDAR sensors by emitting pulsed light waves into the surrounding environment and then analyzing the time difference of receiving the bounced-back pulses. The newly generated orderless point $P_i = \{(x, y, z, r)\}$ is represented by $xyz$ spatial coordinates and reflectance $r$. Based on the point cloud data, ground-truth bounding boxes can be labeled as $B_i = \{b_i\}_{i=1}^{N_i}$ where $b_i \in \mathbb{R}^7$ which include the relative center $xyz$ spatial coordinates to the object ground planes, the box size, the heading angle, and the box label, with their associated bounding box semantic labels $C_i = \{c_i\}_{i=1}^{N_i}$. $N_i$ represents the number of bounding boxes in the $i$-th point cloud. In 3D object detection, a 3D objector $M_\theta$ with parameters $\theta$ first extracts logits $f_i$ from the raw points. These logits are then processed, resulting in modified logits $f_i'$, which are subsequently used by a classification head for predicting the semantic label. Additionally, a regression head will be used to predict the bounding boxes. Finally, the output of the 3D detector is the predicted bounding box $\{b_i'\}_{i=1}^{N_i}$, which includes the semantic labels $\{c_i'\}_{i=1}^{N_i}$ for each bounding box, where $c_i' \in \{1, 2, \ldots, C\}$ with $C$ being the total number of semantic classes, e.g., cars, cyclists and pedestrians.

**Active Learning for 3D Object Detection.** Active 3D object detection aims to reduce the labeling cost for training the 3D detector. In the beginning stage, a small number of labeled point clouds $D_L$ are randomly selected from the unlabeled data pool $D_U$ to train the backbone 3D object detection model. During active learning, for each query iteration $q \in \{1, 2, \ldots, Q\}$, a given active learning method will select $\Gamma$ number of unlabeled point clouds from $D_U$. These selected point clouds are then given to a human annotator for labeling the bounded boxes and semantic labels, and the labeled point clouds $D_S = \{P_j, B_j\}_{j \in [N_q]}$ along with the previously selected point clouds are combined to create selected point clouds $D_L = D_L \cup D_S$ for the backbone model to re-train. When the query round $Q$ is reached or the total queried bounding box number, or budget, $N_Q$ is reached, it will stop repeating the above process, where $N_Q = \sum_{q=1}^{Q} N_q$.

**Submodular Functions and Optimization.** A set function $f$, in the discrete space, defined as follows: $f : 2^D \to \mathbb{R}$, where $2^D$ is a power set of $D$ with $f$ coming from the discrete space in $\mathbb{R}^{2^n}$, is considered submodular if it satisfies the following property of diminishing returns,

$$f(A) + f(B) \geq f(A \cup B) + f(A \cap B) \tag{1}$$

$\forall A, B \subseteq D$. A similar alternative property is,

$$f(A \cup \{x\}) - f(A) \geq f(B \cup \{x\}) - f(B) \tag{2}$$

$\forall A, B \subseteq D$, $A \subseteq B$ and $x \notin B$ [2]. Additionally, $f$ is strictly monotone if $f(A) < f(B)$ for $A \subseteq B$. An important example of a submodular function, in the context of active learning, is the Shannon entropy [48]. We refer the reader to [2] for proof of the submodularity of the Shannon entropy [48].

In order to maximize the representativeness of samples or subsets, a natural solution is to maximize $f$ in the form of $\max_{A \in \mathcal{S}} f(A)$ with $\mathcal{S}$ being a constrained set and $\mathcal{S} \subseteq 2^D$. The idea is to select a subset set that would accept feasible solutions. Since $f$ is monotone submodular, such discrete maximization problems, being NP-complete [10], can be guaranteed to be approximated to a factor of $1 - \frac{1}{e} \approx 0.63$ [37] if maximized under $\mathcal{S}$ using a greedy approximation algorithm [2].

## 4 Algorithms

### 4.1 A Submodular Optimization Approach to Active 3D Object Detection

To tackle the core challenges of active 3D object detection, we propose a **unified** framework grounded in submodular optimization. Our objective is to select a set of unlabeled point clouds that are (1)

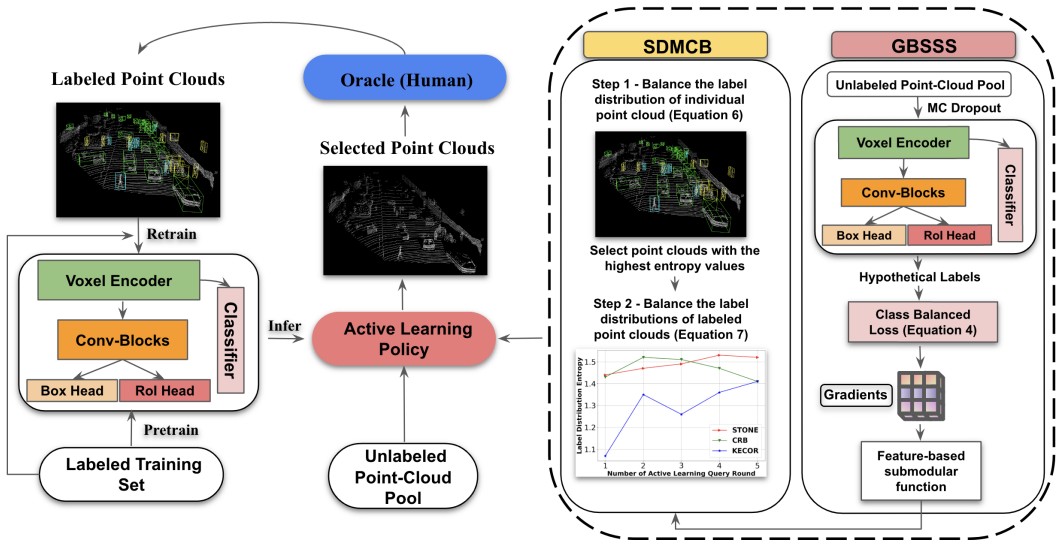

Figure 1: **STONE**: An illustrative pipeline of our proposed active learning method for 3D object detection leveraging submodular functions.

representative, including various levels of difficulty, and, (2) preservative of the label distribution of the selected samples.

To achieve the first goal, we leverage a submodular function $f_1$ to measure the representativeness of the selected unlabeled point clouds $D_S$ with respect to the whole unlabeled set $D_U$. This can be achieved by minimizing the absolute difference between $f_1(D_U)$ and $f_1(D_S)$ since $D_S \subset D_U$. This translates to maximizing the absolute difference $f_1(D_S) - f_1(D_U)$. To achieve the second goal, we leverage a different submodular function $f_2$ to ensure that once the selected unlabeled point clouds $D_S$ are added to the labeled set $D_L$, the overall quality, as measured by label distribution, will not decrease. In sum, we aim to select unlabeled point clouds $D_S$ from the unlabeled pool $D_U$ that optimize the following two objectives,

$$\max_{D_S \subset D_U} [f_1(D_S) - f_1(D_U)] + [f_2(D_L) - f_2(D_L \cup D_S)] \tag{3}$$

Unlike existing active learning methods that rely on sample uncertainty, such as Shannon entropy [48, 55], which tends to select difficult point clouds, our proposed formulation selects samples of varying difficulty levels. Additionally, unlike existing algorithms [44, 18] or clustering-based approaches [38, 57, 3], our formulation also considers the selected labeled point clouds to prevent data imbalance when training a 3D detector. The formulation is also general, allowing for different choices of submodular functions. In practice, we use a feature-based submodular function [59, 2] as $f_1$ and the entropy of the label distribution as $f_2$. Although the proposed formulation is motivated by the problem of active 3D object detection, it can also be useful for active learning problems in similar domains, such as 2D object detection, as we will demonstrate in the experiments. Finding the set $D_S$ that solves Equation 3 is NP-complete [10]. Therefore, we propose a simple algorithm called **STONE** to efficiently solve the problem as will be detailed below.

## 4.2 STONE

In order to optimize the submodular optimization framework in Equation 3, we have designed an active learning pipeline with two stages, aligning with the dual objectives of the submodular criteria. To enhance computational efficiency and simplify optimization, we have implemented a hierarchical structure to eliminate unselected samples at each stage, with the remaining samples comprising our final selection. Initially, we select $\Gamma_1$ samples using the proposed Gradient-Based Submodular Subset Selection (**GBSSS**), which maximizes diversity and coverage while minimizing redundancy from $D_U$. Subsequently, we select $\Gamma_2$ samples from $\Gamma_1$ through Submodular Optimization for Class Balancing (**SDMCB**). A detailed explanation of these stages is provided in the following paragraphs. We illustrate our proposed pipeline in Figure 1.

### 4.2.1 Gradient-Based Submodular Subset Selection (GBSSS)

To address the challenge outlined in the introduction—arising from variations in object size, occlusion, and truncation, which introduce different levels of difficulty across categories—we need to identify representative and diverse samples not only across categories but also within each category with varying difficulty levels. This challenge can be addressed by using a submodular function $f_1$ for maximizing $f_1(D_S) - f_1(D_U)$. In the proposed gradient-based submodular subset selection, we select a subset of unlabeled point clouds from $D_U$ that maximizes gradient feature coverage while ensuring diversity.

Due to the absence of ground-truth semantic labels in the pool of unlabeled point clouds, we employ Monte Carlo dropout (MC-dropout) [13] at the detector head for each point cloud $P_i$ to generate multiple regression predictions, as in [35]. By averaging these predictions, we obtain regression hypothetical labels as well as the classification hypothetical labels. We then calculate the loss function $\mathcal{L}_i$ of the 3D detector $M_\theta$, with parameters $\theta$, using the classification and regression hypothetical labels. Following backpropagation, we extract gradients $\nabla_\theta \mathcal{L}_i$ from the fully connected (FC) layer of the detector head.

However, when the dataset is highly imbalanced, as is usually the case in 3D object detection, the gradients generated by the above approach become inaccurate for classes with fewer samples [39]. To address this issue, we introduce two novel reweighing approaches for handling regression loss $L_{reg}$ and classification loss $L_{cls}$ in 3D object detection for a better computation of the gradients for rare classes. In particular, after calculating the regression loss for each bounding box, we calculate the average loss $L_{reg}^c$ for class $c$ based on the regression hypothetical labels. For a given label $c$, the regression loss reweighing factors can be expressed as $w_c = \frac{1}{n_c}$, where $n_c$ is the number of the bounding boxes that belong to class $c$. We then normalize $w_c$ as $\tilde{w}_c = \frac{w_c}{\max(w_c)}$. The purpose of doing so is two-fold - 1) The class-specific weights in $L_{reg}^c$ are uniformly scaled to prevent bias toward frequently occurring classes. 2) This approach helps reduce overfitting to bounding boxes associated with those classes. Then, we perform an element-wise multiplication between $\tilde{w}_c$ and $L_{reg}^c$ to re-scale the loss, for each class $c$, and compute the mean, across all the semantic classes $C$, to get the final reweighed regression loss $\hat{L}_{reg}$, as $\hat{L}_{reg} = \frac{1}{C}\sum_{c=1}^{C} \tilde{w}_c \cdot L_{reg}^c$. The reweighed regression loss $\hat{L}_{reg}$ places more emphasis on classes with fewer samples, which is critical for computing the gradient.

Inspired by the theoretical formulation in [4], we additionally introduce a new classification reweighing loss function designed to handle class imbalances by adapting the classification margins according to the label distribution during the active learning stage. In essence, for rare classes, the distance of the samples from the decision boundary i.e., the margin, would have to be penalized more. This way the generalization error of minor classes can be improved without worsening the performance on frequent classes. However, penalizing the rare classes more might affect the margins of frequent classes, leading to a complex trade-off [4]. In order to balance this, for the $i$-th point cloud, we leverage a margin vector $m_i$, where the margin for class $c$ is defined as $m_{i,c} = \frac{1}{\sqrt{n_c}}$ ($n_c$ is the class frequency of class $c$). We then subtract the predicted logits $f_i$ by the margin vector $m_i$ to reweigh the predicted logits. Finally, the reweighed logits and hypothetical classification labels $\hat{y}_i$ are used in the classification loss function $L_{cls}$ to compute the class-balanced classification loss $\hat{L}_{cls}$ as,

$$\hat{L}_{cls} = L_{cls}(\hat{y}_i, f_i - m_i) \tag{4}$$

We then use the class-balanced detection loss $\hat{L} = \hat{L}_{reg} + \hat{L}_{cls}$ to compute the gradient for each point cloud. After obtaining the gradients, we use a feature-based submodular function [59] $f_1$ to select the top $\Gamma_1$ diverse samples from $D_U$,

$$\max_{D_S \subset D_U, |D_S| = \Gamma_1} \sum_{P_i \in D_S} g\left(\mu(\nabla_\theta \hat{L}_i)\right) \tag{5}$$

where $g(x) = \log(1 + x)$. The concavity of $g(x)$ ensures that the marginal gain of adding more instances of similar features decreases as more such instances are added. This means that the function will favor adding elements that introduce new features rather than redundant ones. For the score function [59], $\mu(\cdot)$, we use gradients distribution entropy $H(\nabla_\theta \mathcal{L})$ as the measure of informativeness. More details are given in the Appendix. A greedy algorithm is then applied to iteratively add samples to a subset, providing the most significant increase in marginal gain until the target size, $\Gamma_1$, is reached.

#### 4.2.2 Submodular Diversity Maximization for Class Balancing (SDMCB)

One of the main challenges for active 3D object detection is that each 3D scene can contain multiple semantic classes, such as cars and cyclists. As the queried point clouds increase, it will inevitably introduce an imbalanced label distribution in $D_L$, eventually resulting in performance drops. This situation is more prominent in autonomous driving datasets. A recent study, CRB, attempts to address this imbalance issue by introducing individual point cloud balancing. However, it cannot guarantee a balanced label distribution in $D_U$ across multiple active learning queries. In Figure 2, we illustrate the imbalance in label distribution entropy caused by existing methods [35, 34], particularly KECOR, where label imbalance begins to increase as the number of query rounds rises. Aiming to alleviate this label

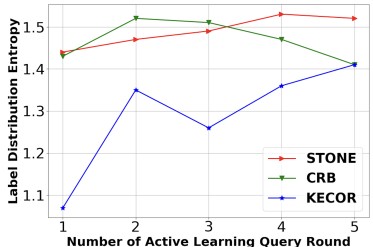

Figure 2: The proposed **STONE** method more effectively maintains the balance of label distribution in active 3D object detection. The plots show the cumulative label distribution entropy values on KITTI [15] validation set split with PV-RCNN [49].

imbalance, we utilize entropy calculated from the predicted label distribution as the submodular function $f_2$. This approach supports the principle of diminishing returns as $D_L$ becomes more balanced. Our method is divided into two steps. In Step 1, we select point clouds which have balanced label distribution based on the hypothetical labels. In Step 2, from the selected ones, we further choose a subset that can be labeled to ensure that the label distribution is balanced in the labeled set $D_L$.

**Step 1: Balance the label distribution of individual point cloud.** To balance each individual point cloud, we select the top $\mathcal{K}$ most balanced samples according to their individual point cloud label distribution entropy. This selection process applies a heuristic that maximizes the immediate gain in entropy. To calculate the probability $\boldsymbol{p}_{i,c}$ of a specific label $c$ within point cloud $P_i$, we first normalize the count of predicted label $c$, denoted as $n_c$, by the total number of predicted bounding boxes $N_i$ in $P_i$. This normalization is done using a softmax function and adjusts the raw count of label $c$ to reflect its proportion relative to the total number of labels. Next, the entropy of the individual point cloud, $H(P_i)$, is calculated as follows,

$$H(P_i) = -\sum_{c=1}^{C} \boldsymbol{p}_{i,c} \log \boldsymbol{p}_{i,c}, \quad \boldsymbol{p}_{i,c} = \frac{e^{n_c / N_i}}{\sum_{c=1}^{C} e^{n_c / N_i}} \tag{6}$$

We calculate $H(P_i)$ for each unlabeled point cloud $P_i \in D_U$ and identify the top $\mathcal{K}_1$ point cloud samples with the highest entropy values.

**Step 2: Balance the label distribution of labeled point clouds.** We then aim to choose a subset of point clouds from the selected ones in Step 1 for balancing the label distribution of the labeled point clouds. To achieve this, we follow a similar approach from the previous step to calculate label probability. Instead of normalizing $n_c$ by the number of bounding boxes $N_i$ in the point cloud $P_i$, we first compute the sum of the number of labeled bounding boxes $N_{L,c}$ of a certain class $c$ in the labeled set and $n_c$, which represents the total number of objects belonging to class $c$ once the point cloud $P_i$ is labeled. Then we compute the sum of total labeled bounding boxes $N_L$ and $N_i$ to find $\gamma_{i,c}$ which represents the normalized number of predicted label $c$. Finally, we calculate $\tilde{\boldsymbol{p}}_{i,c}$ indicating the cumulative label probability of certain class $c$ once the point cloud $P_i$ is labeled. The cumulative entropy $H(\tilde{P}_i)$ of the cumulative label probability then can be computed as,

$$H(\tilde{P}_i) = -\sum_{c=1}^{C} \tilde{\boldsymbol{p}}_{i,c} \log \tilde{\boldsymbol{p}}_{i,c}, \quad \tilde{\boldsymbol{p}}_{i,c} = \frac{e^{\gamma_{i,c}}}{\sum_{c=1}^{C} e^{\gamma_{i,c}}}, \quad \gamma_{i,c} = \frac{N_{L,c} + n_c}{N_L + N_i} \tag{7}$$

$H(\tilde{P}_i)$ can be used to measure label balancing after the point cloud $P_i$ is added into the labeled set. To construct the final selected point clouds in the current query round $q$, we use a greedy search algorithm to iteratively select samples from $D_U$ that maximizes the cumulative entropy, to finally select the top $\Gamma_2$ samples. Thus, using this two-stage progressive approach, we achieve both individual point cloud label balance and label balance across all the labeled point clouds which is crucial for the real-world application of active 3D object detection.

# 5 Experiments and Results

## 5.1 Experimental Setup

**Datasets.** For our experiments, we use the KITTI dataset [15], one of the commonly used datasets in autonomous driving tasks. The dataset consists of 3,712 training samples and 3,769 validation samples, which include a total of 80,256 labeled objects. These objects include cars, pedestrians, and cyclists, each annotated with class categories and bounding boxes. We also use the more challenging dataset for 3D object detection in autonomous driving - the Waymo Open dataset [52]. It includes 158,361 training samples and 40,077 testing samples. In the KITTI dataset, task difficulty levels are defined as EASY for fully visible objects, MODERATE for partially occluded objects, and HARD for significantly occluded objects. For the Waymo test set, the framework categorizes difficulty into two levels: LEVEL 1 with more than five LiDAR points inside the ground-truth bounding box and LEVEL 2 with less or equal to five points. The sampling intervals are set to 1 and 10 for KITTI and Waymo Open, respectively.

**Baselines.** We compare our work against several generic active learning baselines, 1) **RANDOM:** Naive sampling strategy that randomly selects fixed samples at every round; 2) **ENTROPY** [55, 41]: Selects samples with the highest degree of uncertainty as measured by entropy of the sample's posterior probability; 3) **LLAL** [62]: Task-agnostic method with a parametric module that chooses samples where the model is likely to make wrong predictions, based on an indicative loss; 4) **CORESET** [44]: Greedy furthest-first method using the core-set selection on both labeled and unlabeled embeddings. 5) **BADGE** [1]: Batch-mode AL method to select diverse samples, with high gradient magnitude, that span a wide range of directions in the gradient space.

We also draw contrasts between our method and AL methods (and variants) for 2D/3D detection, 6) **MC-MI** [11]: Uses MC-dropout [14] to estimate model uncertainty and mutual information to select the most uncertain point cloud samples; 7) **MC-REG** [35]: Uses several rounds of MC-dropout [14] to approximate the regression uncertainty and picks samples with the greatest variance for labeling; 8) **LT/C** [22]: Adapted from 2D detection, it selects samples based on both localization uncertainty and classification confidence to improve model performance; 9) **CONSENSUS** [43]: Employs ensemble-based uncertainty estimation and continuous training to reduce labeling efforts. 10) **CRB** [35] and 11) **KECOR** [34]. To compare the performance of our method on the 2D object detection task, we compare with 12) **AL-MDN** [5]: Constructs mixture density networks to estimate probability distributions for the outputs of localization and classification heads.

**Evaluation Metrics.** To maintain fairness with the baselines on KITTI, we measure the performance using Average Precision (AP) for 3D and Bird Eye View (BEV) detection, with rotated Intersection over Union (IoU) thresholds of 0.7 for cars and 0.5 for pedestrians and cyclists, following [49]. For the Waymo Open, performance is measured using Average Precision (AP) and Average Precision Weighted by Heading (APH), with IoU thresholds of 0.7 for vehicles and 0.5 for pedestrians and cyclists.

**Implementation Details.** We train our proposed method and all baselines on a GPU cluster with 4 NVIDIA RTX A5000 GPUs. To make fair comparisons, we adopt the implementation settings as outlined in CRB. All the baselines use PV-RCNN [49] as the backbone detection model.

- Training settings. For KITTI and Waymo Open, the training batch sizes are set to 6 and 4 respectively. However, the evaluation batch sizes are set to 16 for both datasets. We optimize the network parameters using Adam with a fixed learning rate of 0.01. For all the methods, we perform 5 stochastic forward passes of the MC-Dropout [14].

- Active learning parameters. For KITTI, we set $\Gamma_1$ and $\Gamma_2$ to 400 and 300 respectively. In the case of Waymo Open, $\Gamma_1$ and $\Gamma_2$ are set to 2,000 and 1,200 respectively. To ensure fairness, $N_q$ is set to 100 in all the methods.

## 5.2 Results

**KITTI Dataset Results.** We evaluate the performance of STONE against other baseline methods in Table 1. We clearly notice that STONE outperforms all the prior active learning methods, irrespective of the detection difficulty level and backbone model. In particular with PV-RCNN [49] as backbone, on average, we observe 3D AP improvements of 1.47%, 0.84%, and 1.24%, over CRB for the EASY,

Table 1: 3D AP(%) scores on KITTI validation set with 1% queried bounding boxes, using PV-RCNN as the backbone detection model.

| | CAR | | | Pedestrian | | | Cyclist | | | Average | | |
| --- | --- | --- | --- | --- | --- | --- | --- | --- | --- | --- | --- | --- |
| Method | EASY | MOD. | HARD | EASY | MOD. | HARD | EASY | MOD. | HARD | EASY | MOD. | HARD |
| CORESET [44] | 87.77 | 77.73 | 72.95 | 47.27 | 41.97 | 38.19 | 81.73 | 59.72 | 55.64 | 72.26 | 59.81 | 55.59 |
| BADGE [1] | 89.96 | 75.78 | 70.54 | 51.94 | 46.24 | 40.98 | 84.11 | 62.29 | 58.12 | 75.34 | 61.44 | 65.55 |
| LLAL [62] | 89.95 | 78.65 | 75.32 | 56.34 | 49.87 | 45.97 | 75.55 | 60.35 | 55.36 | 73.94 | 62.95 | 58.88 |
| MC-REG [35] | 88.85 | 76.21 | 73.47 | 35.82 | 31.81 | 29.79 | 73.98 | 55.23 | 51.85 | 66.21 | 54.41 | 51.70 |
| MC-MI [11] | 86.28 | 75.58 | 71.56 | 41.05 | 37.50 | 33.83 | 86.26 | 60.22 | 56.04 | 71.19 | 57.77 | 53.81 |
| CONSENSUS [43] | 90.14 | 78.01 | 74.28 | 56.43 | 49.50 | 44.80 | 78.46 | 55.77 | 53.73 | 75.01 | 61.09 | 57.60 |
| LT/C [22] | 88.73 | 78.12 | 73.87 | 55.17 | 48.37 | 43.63 | 83.72 | 63.21 | 59.16 | 75.88 | 63.23 | 58.89 |
| CRB [35] | 90.98 | 79.02 | 74.04 | 64.17 | 54.80 | 50.82 | 86.96 | 67.45 | 63.56 | 80.70 | 67.81 | 62.81 |
| KECOR [34] | 91.71 | 79.56 | 74.05 | 65.37 | 57.33 | 51.56 | 87.80 | **69.13** | **64.65** | 81.63 | 68.67 | 63.42 |
| **STONE** | **92.09** | **80.27** | **75.44** | **66.1** | **58.84** | **52.70** | **88.31** | 67.14 | 64.01 | **82.17** | **68.75** | **64.05** |

`MODERATE`, and `HARD` evaluation modes. We observe improvements of 0.54%, 0.08%, and 0.63% over KECOR respectively.

From Figure 3, it is evident that regardless of the detection difficulty level, STONE consistently surpasses other baseline methods. Specifically, compared to the previous state-of-the-art methods KECOR and CRB, with 1% of labeled bounding boxes from the entire pool of unlabeled point clouds at the `HARD` difficulty level, STONE is on average 0.9% higher than KECOR and 1.9% higher than CRB. Although STONE requires 37% more annotated bounding boxes than KECOR, it achieves higher accuracy in earlier query rounds, resulting in faster training times due to fewer epochs and active queries needed. Compared to CRB, even with 36% more labeled bounding boxes at the final round of active queries, STONE is still on average 0.84% higher across all the difficulty levels.

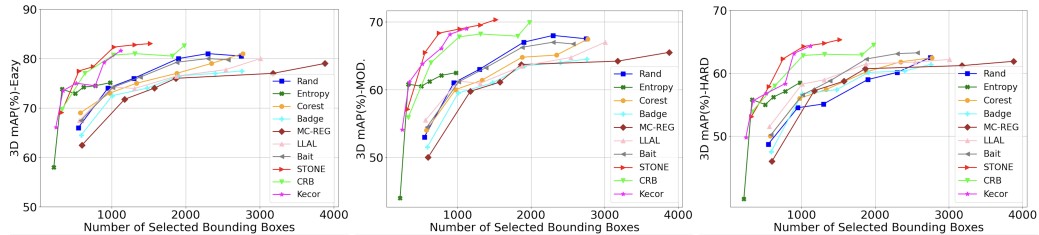

Figure 3: 3D mAP (%) of AL baselines on the KITTI validation set with PV-RCNN.

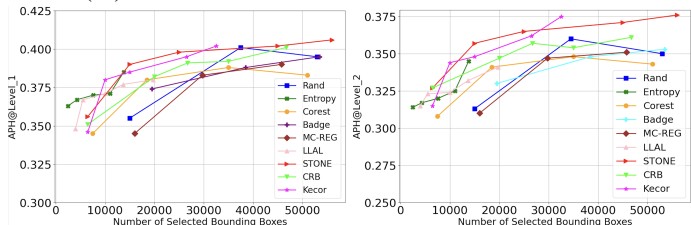

Figure 4: 3D mAP (%) of AL baselines on the Waymo Open validation set with PV-RCNN.

**Waymo Open Dataset Results.** To further test the generality and robustness of STONE, we also evaluate our approach on the Waymo Open dataset using the APH score as the performance metric across the two levels of difficulty `LEVEL 1` and `LEVEL 2`. As shown in Figure 4, STONE surpasses other baseline methods as the number of bounding boxes increases. Compared to previous state-of-the-art methods KECOR and CRB at 25,000 bounding boxes, STONE achieves higher accuracy for both levels of detection difficulty. Our method uses fewer active learning loops than KECOR, meaning it reaches a 2% higher APH score with fewer training epochs. Additionally, when comparing STONE with CRB in the final active learning round, STONE achieves a 1% higher APH score on average with 24% fewer bounding boxes.

**2D Object Detection Results.** We also demonstrate the generalizability of our method on a 2D object detection task. We perform experiments on the PASCAL VOC [9] dataset, that contains 20 object

classes. For the experiments, we use VOC07 *trainval* and VOC07+12 *trainval* to train a Single Shot MultiBox Detector (SSD) [33] with a VGG-16 backbone [50], and test on VOC07 *test*. We follow the training guidelines and setup as outlined in AL-MDN [5] to reproduce the results. For all methods, $N_q$ is set to be 1000 for a total of three queries. In Table 2, we summarize the results in terms of mean Average Precision (mAP), noting that our method either outperforms or is comparable to all baseline methods. In the first query, we observe improvements of 2.93% and 2.43% over AL-MDN$_{gmm}$ and AL-MDN$_{eff}$ respectively. Our method, while achieving a balance in label distribution, also scales well to large datasets with more semantic labels.

**Computational Complexity.** STONE achieves significant GPU memory savings compared to KECOR, which computes gradients of the output of the ROI head's fully connected shared layer, resulting in a gradient matrix of high dimensions and memory. In contrast, STONE focuses gradient computation on the outputs of the classification and regression loss layers within the ROI head, which are much lower in dimensionality. On the KITTI dataset, with a batch size of 6 (for a fair comparison), our method consumes 10 GB of GPU memory, whereas KECOR consumes 24 GB, which is 140% more GPU memory. Additionally, STONE maintains a similar running time to KECOR.

Table 2: VOC07 [9]: mAP(%) of STONE against AL baselines.

| Method | mAP in % (# images) | | |
|---|---|---|---|
| | 1st (2k) | 2nd (3k) | 3rd (4k) |
| RANDOM [33] | 62.43±0.10 | 66.36±0.13 | 68.47±0.09 |
| ENTROPY [41] | 62.43±0.10 | 66.85±0.12 | 68.70±0.18 |
| CORESET [44] | 62.43±0.10 | 66.57±0.20 | 68.57±0.26 |
| LLAL [62] | 62.47±0.16 | 67.02±0.11 | 68.90±0.15 |
| MC-DROPOUT [11] | 62.43±0.19 | 67.10±0.07 | 69.39±0.09 |
| ENSEMBLE [19] | 62.43±0.10 | 67.11±0.26 | 69.26±0.14 |
| AL-MDN$_{gmm}$ [5] | 62.43±0.10 | 67.32±0.12 | 69.43±0.11 |
| AL-MDN$_{eff}$ [5] | 62.91±0.16 | **67.61±0.17** | **69.66±0.17** |
| **STONE** | **65.34±0.34** | 67.01±0.47 | 69.03±0.55 |

Table 3: Ablation on the reweighing factor.

| Components | 3D AP in % | | |
|---|---|---|---|
| | EASY | MOD. | HARD |
| w/o reweighing factor | 80.81 | 68.78 | 62.88 |
| w/o reweighing factor on $L_{reg}$ | 80.32 | 69.64 | 63.59 |
| w/o reweighing factor on $L_{cls}$ | 80.22 | 68.08 | 62.93 |
| none | **83.03** | **70.33** | **65.33** |

Table 4: Stage-wise performance comparisons.

| Stages | 3D AP in % | | | Bounding Boxes |
|---|---|---|---|---|
| | EASY | MOD. | HARD | |
| GBSSS | 79.60 | 66.50 | 62.78 | 2623 |
| GBSSS + SDMCB (Step 1) | 80.37 | 67.80 | 63.17 | 1571 |
| GBSSS + SDMCB (Step 2) | 79.30 | 68.86 | 64.63 | 2473 |
| SDMCB | 80.13 | 67.83 | 62.69 | **1484** |
| SDMCB (Step 1) | 80.98 | 60.44 | 64.31 | 1567 |
| SDMCB (Step 2) | 80.66 | 63.36 | 64.5 | 2514 |
| All | **83.03** | **70.33** | **65.33** | 1530 |

# 6 Ablation Study

We perform ablation studies, on KITTI, to assess the efficacy of our approach and to better understand the key mechanisms and components involved. In the ablation study, all the experiments use 5 rounds of active learning selection, acquiring a total of 500 point clouds for annotation. We present and discuss additional ablation experiments in the Appendix (8).

**Contributions of $\hat{L}_{reg}$ and $\hat{L}_{cls}$.** Given that our method incorporates both regression loss $\hat{L}_{reg}$ and classification loss $\hat{L}_{cls}$ to handle potential class imbalance, we have carried out extensive experiments to analyze their impact on model performance. We first investigated whether regression or classification loss has a greater impact by using gradients generated from each loss separately during the active learning stage. The results showed that using only $\hat{L}_{reg}$ resulted in an average 3D AP drop of 0.77% for HARD while using only $\hat{L}_{cls}$ resulted in an average 3D AP drop of 3.12% for HARD on the KITTI validation dataset. This indicates that $\hat{L}_{cls}$ has a larger impact on our model's performance, and both losses are essential for optimal results.

**Effect of the reweighing factor.** To further evaluate the reweighing factor's importance on our method's performance, we perform three crucial experiments. In Table 3 (row 1), we study the effect of not having the reweighing factor on both the regression and classification losses. We observe performance drops of 3.78%, 1.55%, and 2.45% for the EASY, MODERATE, and HARD levels of difficulty, respectively. We also observe similar performance drops in Table 3 (row 2) and Table 3 (row 3) with no reweighing factor for the regression or classification loss, respectively. This fundamental issue arises because, in contrast to classification tasks where a single loss function is typically considered, detection tasks require the simultaneous optimization of multiple loss components. Specifically, reweighing only the classification or regression loss in detection tasks can disrupt the

Table 5: 3D mAP(%) scores on KITTI validation set with 1% queried bounding boxes, using SECOND [61] as the backbone detection model.

| Method | 3D Detection average mAP | | | BEV Detection average mAP | | |
|---|---|---|---|---|---|---|
| | EASY | MOD. | HARD | EASY | MOD. | HARD |
| Random | 66.33 | 55.48 | 51.53 | 75.66 | 63.77 | 59.71 |
| CORESET [44] | 66.86 | 53.22 | 48.97 | 73.08 | 61.03 | 56.95 |
| LLAL [62] | 69.19 | 55.38 | 50.85 | 76.52 | 63.25 | 59.07 |
| BADGE [1] | 69.92 | 55.60 | 51.23 | 76.07 | 63.39 | 59.47 |
| BAIT [21] | 69.45 | 55.61 | 51.25 | 76.04 | 63.49 | 53.40 |
| CRB [35] | 72.33 | 58.06 | 53.09 | 78.84 | 65.82 | 61.25 |
| KECOR [34] | 74.05 | 60.38 | 55.34 | 80.00 | 68.20 | 63.20 |
| **STONE** | **76.86** | **64.04** | **58.75** | **82.14** | **70.82** | **65.68** |

balance within the model, as it does not adequately model the interdependencies between these loss functions, leading to the suboptimal performance of the detector.

**Relevance of each stage.** In Table 4, we examine the importance and the relevance of each stage of our method. We assess the performance using 3D AP (%) and the number of bounding boxes annotated with 500 point cloud selections. From Table 4 we observe that the removal of any single component leads to a drop in 3D AP performance. In particular, using the GBMSS stage only leads to a drop of 3D AP by 3.43%, 3.83%, and 2.55% for the EASY, MODERATE and HARD difficulty levels respectively. In such a scenario, the bounding boxes annotated increase by 71.43%, which shoots up the labeling cost. It is interesting to note that while using only the SDMCB stage results in annotating the lowest bounding boxes, it also leads to a drop in performance. In conclusion, both stages of our method are necessary for the optimal selection of samples, with a lower labeling cost, to ensure the efficiency and accuracy of the proposed active learning method.

**Backbone-agnostic performance.** In all of the experiments in this paper, we use PV-RCNN as the backbone detection model. However, to show the invariance of our method towards a change in the backbone, we perform experiments with SECOND [61], a widely used 3D object detector. The results, as shown in Table 5, indicate that STONE achieves a 3.4% higher 3D mAP score at the HARD level and 2.43% higher mAP score at the HARD level in BEV detection compared to the state-of-the-art method, KECOR. This demonstrates the performance and generality of the proposed approach.

## 7 Conclusion

In this paper, we propose a novel approach called **STONE**, a **unified** active 3D object detection methodology, based on submodular optimization. We provide a robust and compute-efficient solution by proposing a two-stage algorithm that first utilizes a submodular function based on the gradients using the Monte Carlo dropout [13] to select representative point clouds. We then apply a greedy search algorithm to balance the data distribution due to the possibility of having an imbalance across all the labeled point clouds. Extensive experiments on benchmark autonomous driving datasets, including KITTI [15] and Waymo Open [52] datasets, demonstrate the effectiveness and generalization of our method.

**Limitations.** One limitation of the proposed method is that it does not reduce the running time of existing active 3D detection methods, which is primarily due to the large number of unlabeled point clouds. In future work, we will investigate more efficient methods for active 3D object detection.

**Societal Impact.** The proposed method will be important for several real-world applications such as autonomous driving and robotics, by reducing the cost of labeling.

**Acknowledgements.** We would like to thank the anonymous reviewers for their helpful comments. This project was supported by a grant from the University of Texas at Dallas. This work is also supported by the National Science Foundation under Grant Numbers IIS-2106937, a gift from Google Research, an Amazon Research Award, and the Adobe Data Science Research award.

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

## 8 Appendix

### 8.1 Additional Ablation Studies

**Impact of different submodular functions.** We evaluate STONE with different submodular functions used in stage 1 i.e., GBMSS, and report the results in Table 6. Submodular functions such as facility location have demonstrated effectiveness in modeling representativity and diversity [23], as well as in active learning [28, 27]. We employ these functions in our GBMSS study to assess their effectiveness. In terms of performance measured by 3D AP, we observe results similar to those of STONE. However, it comes with an extremely high labeling cost with a 50.98% rise in bounding box annotations. With the max coverage submodular function, we see slight performance drops of 1.57%, 0.06%, and 0.65% for the EASY, MODERATE and HARD difficulty levels respectively. Overall, our method attains the best results in terms of 3D average precision (AP) while maintaining a low labeling budget.

Table 6: Impact of different submodular functions.

| Submodular functions | 3D AP in % | | | Bounding Boxes |
| | EASY | MOD. | HARD | |
| --- | --- | --- | --- | --- |
| Facility location | 82.91 | 70.64 | 65.98 | 3121 |
| Max coverage | 81.46 | 69.47 | 64.68 | 1674 |
| STONE | **83.03** | **70.33** | **65.33** | **1530** |

Table 7: Sensitivity to thresholds $\Gamma_1, \Gamma_2$.

| Components | 3D AP in % | | |
| | EASY | MOD. | HARD |
| --- | --- | --- | --- |
| $\Gamma_1$: 400, $\Gamma_2$: 200 | 82.32 | 68.24 | 64.08 |
| $\Gamma_1$: **400**, $\Gamma_2$: **300** | **83.03** | **70.33** | **65.33** |
| $\Gamma_1$: 500, $\Gamma_2$: 200 | 78.62 | 67.44 | 63.77 |
| $\Gamma_1$: 500, $\Gamma_2$: 200 | 80.70 | 69.18 | 64.01 |
| $\Gamma_1$: 600, $\Gamma_2$: 200 | 82.53 | 68.90 | 63.82 |
| $\Gamma_1$: 600, $\Gamma_2$: 300 | 81.67 | 67.47 | 64.72 |

**Sensitivity to thresholds $\Gamma_1$ and $\Gamma_2$.** We conduct a sensitivity analysis of model performance to different variants of thresholds $\Gamma_1$ and $\Gamma_2$, at all difficulty levels, to understand their importance in our method. We report the 3D AP (in %) across six different combinations in Table 7 with 500 point-cloud selections. For the MODERATE and HARD difficulty levels, we observe fluctuations within 2.89% and 1.56% (compared to the lowest). This suggests that our method is relatively stable with the best $\Gamma_1$ and $\Gamma_2$ being 400 and 300 respectively.

Table 8: Performance comparisons of STONE and AL baselines using 3D AP(%) scores on the KITTI validation set (HARD level) with PV-RCNN as the backbone architecture.

| Method | 3D AP % using 1% (bounding box) | 2% | 3% |
| --- | --- | --- | --- |
| CRB | 62.81 | 65.43 | 69.93 |
| KECOR | 63.42 | 67.25 | 71.70 |
| STONE | 64.05 | 66.83 | 70.86 |

Table 9: Performance comparisons of STONE and AL baselines using 3D AP(%) scores on the KITTI validation set (HARD level) with SECOND [61] as the backbone architecture.

| Method | 3D AP % using 1% (bounding box) | 2% | 3% |
| --- | --- | --- | --- |
| CRB | 53.09 | 55.67 | 57.01 |
| KECOR | 55.34 | 57.56 | 58.92 |
| STONE | 58.75 | 60.33 | 61.89 |

**Effect of % labeled bounding boxes.** To ensure a fair comparison with the previous state-of-the-art methods CRB [35] and KECOR [34], we leveraged 1% of the labeled bounding boxes. Referring to Tables 8 and 9, as more labeled bounding boxes are added to the training, the results get better. It is worth noting that the KECOR method marginally surpasses STONE when using 2% and 3% of the bounding box. This is because the STONE method, when utilizing the PV-RCNN backbone, tends to select scenes with more objects. As a result, STONE reaches the bounding box limit very early in the active learning stage. The slightly better results achieved by KECOR are due to it querying more scenes and being trained over more epochs. In active 3D object detection, the goal is to label as few bounding boxes as possible to achieve good performance. Therefore, it is critical to maintain high performance with a smaller number of labeled bounding boxes, as demonstrated by STONE.

