# OpenReview forum: "STONE: A Submodular Optimization Framework for Active 3D Object Detection"
_NeurIPS.cc/2024/Conference — NeurIPS 2024 poster_

### Official Review · Reviewer_qVYS · 2024-07-11

**Soundness:** 3
**Presentation:** 3
**Contribution:** 3
**Rating:** 6
**Confidence:** 3

**Summary:**

The paper proposes an approach to address challenges in active learning for 3d object detection to select unlabeled point clouds for further labeling. The labeling criteria maximises representativeness of the chosen point cloud with respected to an unlabeled point cloud set and also making sure that classes are balanced and have similar entropy as the labeled dataset.
The paper proposes two submodular functions (GBSS, SDMCB) to achieve the criteria mentioned above. They lead to better performance compared to the current SOTA. GBSSS ensures that the selected point clouds are diverse and representative. SDMCB tries to reduce the bias that arises from imbalance in classes in the dataset.

**Strengths:**

S1. Point Cloud Choice. The paper proposes a way to choose point clouds based on varying ranges of difficulty which is useful.

S2. Treating Data Imbalance. The paper addresses data imbalance by choosing the point clouds unlike previous approaches. A novel weighting scheme is proposed in classification and regression losses to avoid ignoring classes that occur less frequently.

S3. Marginal Performance boost. The paper achieves better performance compared to other approaches.

S4. Design Justification. Tha ablation studies clearly indicate the improvement with each proposed stage, reweighting factors, selected loss terms in submodular optimization.

**Weaknesses:**

W1. Performance boost is marginal. Although there is some improvement in the performance, it is marginal.

**Questions:**

Q1. Considering Labeled Data for Optimization. The first submodular function maximises the representative capacity of the chosen point cloud compared to the unlabeled set. Why is already labeled data not being considered in the optimization objective? Is it because the pretrained network is a representative of labeled dataset?

Q2. Difficulty enforcement during active learning. It is not clear how HARD, MODERATE AND EASY is enforced during active learning since the objective function is the same and we do not know about the unlabeled pool of point clouds. Is it possible to categorize unlabeled pointclouds?

Q3. Objective Function. The objective in the equation tries to maximize the term while it is mentioned as minimizing the objective in line 142. This seems a mistake.

=======

Those could be answered during discussion phase. I am therefore raising my rating to "weak accept".

**Limitations:**

Yes, addressed.

---

> ### Author Rebuttal · Authors · 2024-08-06
>
> Thank you for your constructive review and valuable suggestions! Below, we provide a detailed response to your questions and comments. If any of our responses fail to sufficiently address your concerns, please inform us, and we will promptly follow up.
>
> #### **[W1] Performance boost is marginal. Although there is some improvement in the performance, it is marginal.**
> We further use SECOND [1], a widely used 3D object detector, as the base model.  The results indicate that STONE achieves a **3.4%** higher mAP score at the hard level in 3D detection and **2.43%** higher mAP score at the hard level in BEV detection compared to the state-of-the-art method, KECOR, as shown in Table 1. This demonstrates the performance and generality of the proposed approach.
>
> #### Table 1: 3D mAP(\%) of STONE and AL baselines on KITTI validation set with 1\% queried bounding boxes with one-stage 3D detector backbone SECOND
> | **Methods** | **3D Detection mAP EASY \%** | **3D Detection mAP MOD. \%** | **3D Detection mAP HARD \%** | **BEV Detection mAP EASY \%** | **BEV Detection mAP MOD. \%** | **BEV Detection mAP HARD \%** |
> |----------------|------------------------------|------------------------------|------------------------------|------------------------------|------------------------------|------------------------------|
> | RAND | 69.33±0.62 | 55.48±0.42| 51.53±0.33 | 75.66±1.10 | 63.77±0.86  | 57.71±0.95  |
> | CORESET | 66.86±2.27 | 53.22±1.65 | 48.97±1.42 | 73.08±1.80| 61.03±1.98 | 56.95±1.53 |
> | LLAL           | 69.19±3.43                   | 55.38±3.63                   | 50.85±3.24                   | 76.52±2.24                   | 63.25±3.11                   | 59.07±2.80                   |
> | BADGE          | 69.92±2.90                   | 55.60±2.72                   | 51.23±2.58                   | 76.07±2.70                   | 63.39±2.52                   | 59.47±2.49                   |               |
> | CRB            | 72.33±0.35                   | 58.06±0.30                   | 53.09±0.31                   | 78.84±0.27                   | 65.82±0.07                   | 61.25±0.22                   |
> | KECOR          | 74.05±0.16                   | 60.68±0.13                   | 55.34±0.23                   | 80.00±0.12                   | 68.20±0.35                   | 63.26±0.25                   |
> | **STONE**      | **76.86±0.88**               | **64.04±0.27**               | **58.75±0.58**               | **82.14±0.90**               | **70.82±0.14**               | **65.68±0.42**               |
>
> #### **[Q1]  The first submodular function maximises the representative capacity of the chosen point cloud compared to the unlabeled set. Why is already labeled data not being considered in the optimization objective? Is it because the pretrained network is a representative of labeled dataset?**
>
> Thanks for pointing this out. Yes, the pretrained model is representative of the labeled dataset as it has already been trained on the labeled point clouds. Due to this reason, only the unlabeled point clouds are considered as we aim to select unlabeled point clouds that are representative of the whole unlabeled pool which can be further labeled to improve the model performance.
>
> #### **[Q2] Difficulty enforcement during active learning. It is not clear how HARD, MODERATE AND EASY is enforced during active learning since the objective function is the same and we do not know about the unlabeled pool of point clouds. Is it possible to categorize unlabeled point clouds?**
>
> Since we do not have access to the true labels of point clouds in the unlabeled pool, categorizing them by difficulty levels is highly challenging. To address this, we use Monte Carlo dropout to estimate the true labels. These estimated labels are then utilized in Gradient-Based Submodular Subset Selection (GBSSS). By employing a feature-based submodular function, we select representative samples from the unlabeled pool, aiming to cover samples of varying difficulty levels. The results in Table 1 of the paper demonstrate that our method achieves the best performance across all difficulty levels, highlighting the effectiveness of our proposed approach.
>
>
> #### **[Q3] Objective Function. The objective in the equation tries to maximize the term while it is mentioned as minimizing the objective in line 142. This seems a mistake.**
>
> Sorry for the confusion. In line 142, we mentioned that we aim to minimize the absolute difference between $f_1(D_U )$ and $f_1(D_S )$. Since $D_S \subset  D_U$, from the property of submodular function we have $f_1(D_S ) \le f_1(D_U )$. In Equation 3, we maximize $[f_1(D_S) - f_1(D_U)]$, which is equivalent to minimizing the absolute difference between $f_1(D_U )$ and $f_1(D_S )$. We will clarify this further in the final version.
>
> [1] Yan Y, Mao Y, Li B. Second: Sparsely embedded convolutional detection[J]. Sensors, 2018, 18(10): 3337.

---

> > ### Comment · Reviewer_qVYS · 2024-08-09
> >
> > Thanks for the clarifications!
> > All questions from my end are answered and I am happy to slightly increase my rating given these justifications.
> > I'd further encourage the authors to include the details on Q1, Q2 in the paper even more explicitly for others to also easily understand.

---

> > > ### Author Response · Authors · 2024-08-13
> > >
> > > Dear Reviewer,
> > >
> > > We appreciate your insightful and valuable feedback once again. We will be adding more details and explanations in response to questions 1 and 2 to help readers better understand in camera-ready version.
> > >
> > > Thank you for your feedback!
> > >
> > > Best Regards,
> > >
> > > The Authors

---

### Official Review · Reviewer_Lj2A · 2024-07-12

**Soundness:** 2
**Presentation:** 2
**Contribution:** 2
**Rating:** 6
**Confidence:** 4

**Summary:**

The paper proposes a novel submodular-based active learning approach STONE for lidar-based 3D object detection. The method then does data balancing using a greedy search algorithm. The method achieves significant improvements on the KITTI and Waymo datasets.

**Strengths:**

+ The paper proposes submodular optimization which is nice.
+ The paper achieves good improvements on the two datasets.

**Weaknesses:**

- The paper relies solely on PV-RCNN as the baseline detector. Quantitaively benchmarking the results with recent detectors like FocalFormer [A] and PillarNet [B] on datasets would strengthen the evaluation.

- Including quantitative evaluations of baselines and STONE on any two established leaderboards: KITTI, Waymo, or nuScenes would let us know the STONE's generalizability.

- Figure 3 presents performance comparisons up to 50k boxes for the proposed STONE method but only 30k boxes for the baseline Kecor. Including results for Kecor at 50k boxes would provide a fairer comparison.

- The paper focuses on active learning. Could the method be adapted for unlabeled data to detect moving objects? If so, a quantitative comparison against MODEST on KITTI and Lyft datasets would be beneficial.

- Figure 1 could be enhanced by illustrating the entropy after step 1 and step 2 of SDMCB to 10 active learning rounds.

- While the application on the Pascal VOC dataset is interesting, quantitative benchmarking of the MS-COCO dataset with base detectors like Faster R-CNN, YOLOv5, and DETR using ResNet-50/101 backbones would be a more comprehensive evaluation for 2D detection.

- The authors mention that varying degree of difficulty comes from multiple sources. I would, therefore, want to compare the 2D detection performance of a baseline and KECOR on the KITTI-360 dataset [D] which has a heavily-truncated building category [D].

References:
- [A] Chen et al., FocalFormer3D: Focusing on Hard Instance for 3D Object Detection, ICCV 2023

- [B] Shi et al., Pillarnet: Real-time and high-performance pillar-based 3d object detection, ECCV 2022

- [C] You et al., MODEST: Learning to Detect Mobile Objects from LiDAR Scans Without Labels, CVPR 2022

- [D] KITTI-360: A novel dataset and benchmarks for urban scene understanding in 2D and 3D, Liao et al., TPAMI 2022.

**Questions:**

Please see the weaknesses.

**Limitations:**

Yes

---

> ### Author Rebuttal · Authors · 2024-08-06
>
> Thank you for your constructive review and valuable suggestions! Below, we provide a detailed response to your questions and comments.
> #### **[W1] Quantitaively benchmarking the results with recent detectors like FocalFormer and PillarNet on datasets would strengthen the evaluation.**
> We evaluated our method alongside other AL baselines using PillarNet (Table 1), and we also implemented the one-stage detector SECOND [1] to draw comprehensive and robust conclusions. Our method consistently outperformed other baselines by a significant margin. The results can be viewed in Table 1 and Table 2.
> #### Table 1: 3D mAP(%) of STONE and AL baselines on KITTI with PillarNet
> | Components | 3D Detection mAP EASY % | MOD. | HARD | BEV Detection mAP EASY % | MOD.| HARD |
> |-|-|-|-|-|-|-|
> | KECOR|28.98|24.79|23.71|31.00| 28.67| 27.26|
> | **STONE**| **29.42** | **27.09**| **24.35** | **32.14** | **30.12** | **29.23** |
> #### Table 2: 3D mAP(\%) of STONE and AL baselines on KITTI with backbone SECOND
> | Methods | 3D Detection mAP EASY \% |MOD.|HARD|BEV Detection mAP EASY \%| MOD.|HARD|
> |-|-|-|-|-|-|--|
> |KECOR|74.05|60.68|55.34|80.00|68.20|63.26|
> |**STONE**|**76.86**|**64.04**| **58.75** | **82.14** | **70.82**  | **65.68** |
>
> #### **[W2] Including quantitative evaluations of baselines and STONE on any two established leaderboards: KITTI, Waymo, or nuScenes would let us know the STONE's generalizability.**
> With a more challenging dataset like nuScenes, which has 10 classes compared to KITTI and Waymo, both of which have 3 classes, our proposed method still demonstrates a performance advantage, showing that it has good generalizability.
> #### Table 3: Performance comparisons of STONE and AL baselines on the NuScenes validation set with SECOND as the backbone architecture.
>
> |Method|nuScenes detection score (NDS)|mAP|
> |-|-|-|
> |Random|37.25|28.96|
> |KECOR|46.92|40.23|
> | **STONE** | **48.79** | **41.23** |
> #### **[W3] Figure 3 presents performance comparisons up to 50k boxes for the proposed STONE method but only 30k boxes for the baseline Kecor. Including results for Kecor at 50k boxes would provide a fairer comparison.**
> KECOR method slightly outperforms the STONE method with 30k, 40k, and 50k bounding boxes (Table 4) because it queries more scenes and undergoes more training epochs. In contrast, STONE, when using the PV-RCNN backbone, tends to reach its bounding box limit early due to its preference for object-rich scenes. In active 3D object detection, the goal is to label as few bounding boxes as possible to achieve good performance. Therefore, it is critical to maintain high performance with a small number of labeled bounding boxes, as demonstrated by STONE.  With the SECOND backbone, STONE consistently outperforms other active learning baselines (Table 5).
>
> #### Table 4: Performance comparisons of STONE and AL baselines using 3D AP(%) scores on the Waymo with PV-RCNN as the backbone.
> |Method|20k (bounding box)|30k|40k|50k|
> |-|-|-|-|-|
> | KECOR| 0.358 | **0.371** | **0.382** | **0.396** |
> | **STONE** | **0.361** | 0.365| 0.369| 0.375 |
> #### Table 5: Performance comparisons of STONE and AL baselines using 3D AP(%) scores on the Waymo with SECOND as the backbone.
> |Method|20k (bounding box)|30k|40k|50k|
> |-|-|-|-|-|
> | KECOR| 0.413| 0.439| 0.473| 0.488|
> | **STONE**| **0.428**| **0.457** | **0.489** | **0.503** |
>
> #### **[W4] Could the method be adapted for unlabeled data to detect moving objects? If so, a quantitative comparison against MODEST on KITTI and Lyft datasets would be beneficial.**
> Thanks for pointing this out. The proposed method is a general active learning method for 3D detection and is not specifically designed for detecting moving objects. Additionally, since MODEST does not classify different object types, a direct comparison is difficult.
>
> #### **[W5] Figure 1 could be enhanced by illustrating the entropy after step 1 and step 2 of SDMCB to 10 active learning rounds.**
> The results can be seen in Figure 2 of the attached PDF within the general response. The red line represents the predicted labels after step one, appearing more balanced compared to the entropy results after the SDMCB (step 2). However, since in step one, we only have access to predicted labels, which can be inaccurate, the estimated entropy cannot accurately reflect the label distribution. In step two, having access to the true labels of the previously selected point clouds allows the entropy to better reflect the label distribution. The drop in entropy value is observed because the total number of objects in the KITTI dataset is highly imbalanced (14,385 cars, 893 cyclists, and 2,280 pedestrians). As more cyclists are selected in the first few rounds, fewer are available for the last few rounds.
>
> #### **[W6] Quantitative benchmarking of the MS-COCO dataset with base detectors like Faster R-CNN, YOLOv5, and DETR using ResNet-50/101 backbones for 2D detection.**
> We use Faster R-CNN as the detector and the MS-COCO dataset, and we leverage the codebase from "Plug and Play Active Learning for Object Detection" presented at CVPR 2024 for a fair comparison. The results demonstrate STONE is competitive with KECOR. Critically, STONE is more memory-efficient compared with KECOR which is practically important for resource-constraint settings.
> #### Table 6:Performance comparisons of STONE and AL baselines on MS-COCO with Faster R-CNN as the backbone architecture.
> |Method| 20% (training data)|25%|30%|35%|40%|
> |-|-|-|-|-|-|
> |KECOR| **27.36**|**29.41**| 30.21| 31.09| 31.87|
> |**STONE**| 27.13|29.36| **30.36** | **31.74** |**32.11**|
> #### **[W7] Compare the 2D detection performance of a baseline and KECOR on the KITTI-360 dataset [D] which has a heavily-truncated building category.**
> Due to time and computing resource limitations, we are unable to conduct this experiment, and we will include the results in the final version of the paper.
>
> [1] Yan Y, Mao Y, Li B. Second: Sparsely embedded convolutional detection[J]. Sensors, 2018, 18(10): 3337.

---

> ### Author Response · Authors · 2024-08-12
>
> Dear reviewer,
>
> We sincerely appreciate your valuable feedback, which has greatly improved our paper! We kindly ask if you could confirm whether our response has adequately addressed your concerns. If so, we would be grateful if you might consider raising your rating. Please do not hesitate to let us know if there are any remaining issues.
>
> Thank you once again for your insightful feedback!
>
> Best Regards,
>
> The Authors

---

> > ### Comment · Reviewer_Lj2A · 2024-08-13
> > **Reply to Authors**
> >
> > Thank you authors for putting a strong rebuttal. I change my rating to WA conditional to the following:
> > - Including Faster R-CNN only for 2D detection on MS-COCO is insufficient since it is an old detector. Therefore, the authors should include results of DETR and YOLOv5 on this task in the main paper in the camera-ready version. That will truly reflect the advantages of the proposed method.
> > - The authors also run and put the KITTI-360 results in the main paper in the camera-ready version.

---

> > > ### Author Response · Authors · 2024-08-13
> > >
> > > Dear Reviewer,
> > >
> > > We appreciate your insightful and valuable feedback once again. We will run and present the results of our method using DETR and YOLOv5 for 2D detection on the MS-COCO dataset. We will also continue working on the KITTI-360 dataset for 2D object detection and present our results once we have the results.
> > >
> > > Thank you for your insightful feedback!
> > >
> > > Best Regards,
> > >
> > > The Authors

---

### Official Review · Reviewer_nHfr · 2024-07-12

**Soundness:** 3
**Presentation:** 3
**Contribution:** 3
**Rating:** 6
**Confidence:** 3

**Summary:**

This paper introduces a framework to reduce the labeling costs of 3D point cloud data in 3D object detection by using a submodular optimization approach. It tries to optimize for data imbalance the distribution of the data like the varying difficulty levels. By the combination of a transformer-architecture and an active learning approach I achieves SOTA results. To do so it optimizes two submodular functions. The first one represents the different difficulty levels and the second one ensures class balance.

**Strengths:**

- The paper mentions important related work and describes the necessary background well

- The introduced submodular optimization framework is a novel contribution that can potentially be applied to other domains of active learning as well

- The introduced components are well explained in detail

- The training is described extensively

**Weaknesses:**

- If not intended, in line 113-144 it could be mentioned that the randomly selected number of point clouds D_L in the the beginning is labeled. This could be seen as obvious but it would make the description more logical as the unlabeled point clouds during training need to be annotated by a human annotator as well.

- To allow a complete and fair comparison with other approaches, evaluations on additional datasets like nuScenes and/or TUM Traffic Intersection would be good

- A better explanation of the Figure 2 would be helpful i.e. why is there not the same amount of datapoints for every method and why are the number of bounding boxes so different. Extrapolating some curves would suggest that some methods work better than STONE. However this can not be evaluated as the datapoints are missing

- The claim that the results are SOTA does not sound tenable, insofar as the percentage improvements are only small. In addition, the diagrams indicate that other methods could deliver better results on closer inspection. In Figure 3, it appears that KECOR would provide better results if more data points and more bounding boxes were available. As these data points are missing, a more accurate comparison is not possible. Furthermore, in Table 1, the best values for cyclists in the moderate and hard cases are not highlighted in bold. This was not necessarily intentional, but should be noted

- Some spelling/grammatical/contextual errors (no need to respond):

Line 48: function -> functions

Line 75-78: include BADGE -> like BADGE?

Line 383 mAP -> AP?

**Questions:**

- What about this paper https://arxiv.org/pdf/2402.03235? They also apply active learning on 3D object detection. However, a comparison is not easy as they use the nuScenes and TUM Traffic intersection dataset (see 3rd point of weaknesses).

**Limitations:**

Were discussed.

---

> ### Author Rebuttal · Authors · 2024-08-06
>
> Thank you for your constructive review and valuable suggestions! Below, we provide a detailed response to your questions and comments.
>
> #### **[W1] If not intended, in line 113-144 it could be mentioned that the randomly selected number of point clouds D_L in the the beginning is labeled.**
>
> Thanks for the suggestion! We will make it clear that the randomly selected point clouds, in the beginning, are labeled, in the final version.
>
> #### **[W2] To allow a complete and fair comparison with other approaches, evaluations on additional datasets like nuScenes and/or TUM Traffic Intersection would be good.**
>
> With a more challenging dataset like nuScenes, which has 10 classes compared to KITTI and Waymo, both of which have 3 classes, our proposed method still demonstrates a performance advantage (Table 1), showing that it has good generalizability.
>
> #### Table 1: Performance comparisons of STONE and AL baselines on the NuScenes validation set with SECOND [1] as the backbone architecture.
>
> | **Method** | **nuScenes detection score (NDS)** | **mAP** |
> |--|--|--|
> | Random | 37.25 | 28.96|
> | KECOR | 46.92| 40.23|
> | **STONE** | **48.79** | **41.23** |
>
> #### **[W3] A better explanation of the Figure 2 would be helpful i.e. why is there not the same amount of datapoints for every method and why are the number of bounding boxes so different. Extrapolating some curves would suggest that some methods work better than STONE. However this can not be evaluated as the datapoints are missing**
>
> Due to some point clouds having more objects than others, we first fixed the budget to the maximum number of point clouds that can be queried, using only 1% of the labeled bounding boxes to ensure a fair comparison. This is consistent with previous works in this domain. Referring to Tables 2 and 3, as more labeled bounding boxes are added to the training, the results will improve. It is worth noting that the KECOR method marginally surpasses the STONE method when using 2\% and 3\% of the bounding box. This is because the STONE method, when utilizing the PV-RCNN backbone, tends to select scenes with more objects. As a result, STONE reaches the bounding box limit very early in the active learning stage. The slightly better results achieved by KECOR are due to it querying more scenes and being trained over more epochs. In active 3D object detection, the goal is to label as few bounding boxes as possible to achieve good performance. Therefore, it is critical to maintain high performance with a small number of labelled bounding boxes, as demonstrated by STONE. When using SECOND [1] as the backbone detector, STONE is consistently better than other state-of-the-art AL baselines (Table 3) even when using more bounding boxes.
>
> #### Table 2: Performance comparisons of STONE and AL baselines using 3D AP(%) scores on the KITTI validation set (HARD level) with PV-RCNN as the backbone.
> | Method | 3D AP \% using 1% (bounding box) |2\% |  3\% |
> |-|-|--|-|
> |CRB| 62.8|65.43 | 69.93|
> |KECOR|63.42|**67.25**|**71.70**|
> |**STONE**|**64.05**|66.83| 70.86|
> #### Table 3: Performance comparisons of STONE and AL baselines using 3D AP(%) scores on the KITTI validation set (HARD level) with SECOND [1] as the backbone.
> | Method | 3D AP \% using 1% (bounding box)|2\% |3 \%|
> |-|-|-|-|
> |CRB|53.09|55.67|57.01|
> |KECOR|55.34|57.56|58.92|
> |**STONE**|**58.75**|**60.33**|**61.89**|
> #### **[W4] The claim that the results are SOTA does not sound tenable, insofar as the percentage improvements are only small. In addition, the diagrams indicate that other methods could deliver better results on closer inspection. In Figure 3, it appears that KECOR would provide better results if more data points and more bounding boxes were available. As these data points are missing, a more accurate comparison is not possible. Furthermore, in Table 1, the best values for cyclists in the moderate and hard cases are not highlighted in bold.**
> We further use SECOND [1], a widely used 3D object detector, as the base model.  The results in Table 4 indicate that STONE achieves a **3.4%** higher mAP score at the hard level in 3D detection and **2.43%** higher mAP score at the hard level in BEV detection compared to the state-of-the-art method, KECOR, as shown in Table 4. This demonstrates the performance and generality of the proposed approach. Missing data can be viewed in Tables 2 and 3. We will add the highlights in Table 1 in the final version of the paper.
> #### Table 4: 3D mAP(\%) of STONE and AL baselines on KITTI validation set with 1\% queried bounding boxes with one-stage 3D detector backbone SECOND [1]
> |Methods| **3D Detection mAP EASY \%** | **3D Detection mAP MOD. \%** | **3D Detection mAP HARD \%** | **BEV Detection mAP EASY \%** | **BEV Detection mAP MOD. \%** | **BEV Detection mAP HARD \%** |
> |-|-|-|-|-|-|-|
> |RAND| 69.33±0.62 | 55.48±0.42 | 51.53±0.33 | 75.66±1.10 | 63.77±0.86 | 57.71±0.95|
> |CORESET| 66.86±2.27| 53.22±1.65 | 48.97±1.42 | 73.08±1.80 | 61.03±1.98  | 56.95±1.53 |
> |LLAL  | 69.19±3.43 | 55.38±3.63 | 50.85±3.24 | 76.52±2.24 | 63.25±3.11| 59.07±2.80 |
> |BADGE| 69.92±2.90| 55.60±2.72| 51.23±2.58| 76.07±2.70| 63.39±2.52 | 59.47±2.49|
> |CRB | 72.33±0.35| 58.06±0.30| 53.09±0.31| 78.84±0.27 | 65.82±0.07| 61.25±0.22|
> |KECOR| 74.05±0.16  | 60.68±0.13 | 55.34±0.23 | 80.00±0.12  | 68.20±0.35| 63.26±0.25 |
> | **STONE**| **76.86±0.88**  | **64.04±0.27**  | **58.75±0.58** | **82.14±0.90**  | **70.82±0.14**  | **65.68±0.42** |
>
> #### **[Q1] What about this paper https://arxiv.org/pdf/2402.03235? However, a comparison is not easy as they use the nuScenes and TUM Traffic intersection dataset**
> Please refer to the answer to [W2] for results on NuScenes. In https://arxiv.org/pdf/2402.03235, the authors consider of case of uniform label and point cloud distributions, however, our approach can be applied to real-world imbalanced label distribution which is more practical.
>
> [1] Yan Y, Mao Y, Li B. Second: Sparsely embedded convolutional detection[J]. Sensors, 2018, 18(10): 3337.

---

> > ### Comment · Reviewer_nHfr · 2024-08-12
> >
> > Thank you for the rebuttal and the clarifications!

---

> > > ### Author Response · Authors · 2024-08-13
> > >
> > > Dear Reviewer,
> > >
> > > We appreciate your insightful and valuable feedback once again. We will be adding the above experiment results in the camera-ready version.
> > >
> > > Best Regards,
> > >
> > > The Authors

---

### Official Review · Reviewer_4dDG · 2024-07-12

**Soundness:** 3
**Presentation:** 3
**Contribution:** 3
**Rating:** 6
**Confidence:** 4

**Summary:**

This paper introduces a novel framework to reduce labeling costs in 3D object detection. Using submodular optimization, the framework addresses data imbalance and varying difficulty levels in LiDAR point cloud. It employs a two-stage algorithm: Gradient-Based Submodular Subset Selection (GBSSS) for selecting diverse and representative point clouds, and Submodular Diversity Maximization for Class Balancing (SDMCB) to ensure balanced label distribution. Experiments on KITTI and WOD datasets show that STONE outperforms existing methods, demonstrating high computational efficiency and state-of-the-art performance.

**Strengths:**

- Reduces labeling cost with submodular optimization
- Addresses data imbalance effectively
- Achieves state-of-the-art performance on KITTI and Waymo datasets
- Demonstrates generalizability to both 3D and 2D object detection tasks

**Weaknesses:**

- Generalizability of Submodular Functions: The paper uses specific submodular functions tailored to their problem, but it lacks a detailed analysis of how generalizable these functions are to other datasets or slightly different tasks (e.g., semantic segmentation). This limits the understanding of the robustness of the proposed method across diverse scenarios.
- Handling of Data Imbalance: Although the paper proposes a two-stage algorithm to handle data imbalance, it does not provide a detailed comparison with other state-of-the-art methods specifically addressing data imbalance in 3D object detection. More in-depth comparative analysis would strengthen the claims regarding its effectiveness in balancing label distribution.
- While the paper claims high computational efficiency for STONE, it does not provide specific data on computation time or resource usage. What are the experimental settings and resource usage details?

**Questions:**

While the method demonstrates effectiveness in 3D object detection, its generalizability to other domains or tasks is not sufficiently discussed. Is the STONE method applicable to other types of 3D data or different application scenarios?

**Limitations:**

Limitations have been adequately addressed in the paper.

---

> ### Author Rebuttal · Authors · 2024-08-06
>
> Thank you for your constructive review and valuable suggestions! Below, we provide a detailed response to your questions and comments.
>
> #### **[W1] Generalizability of Submodular Functions: The paper uses specific submodular functions tailored to their problem, but it lacks a detailed analysis of how generalizable these functions are to other datasets or slightly different tasks (e.g., semantic segmentation). This limits the understanding of the robustness of the proposed method across diverse scenarios.**
>
> We implemented our method in the active learning domain for 3D semantic segmentation following [1], utilizing the nuScenes dataset [3] with MinkNet as the backbone. Even though our method is not particularly designed for 3D semantic segmentation, it still achieves better performance compared with traditional active learning methods like Entropy and obtains competitive performance compared to the state-of-the-art 3D semantic segmentation method Annotator [1].  Detailed per-class results are available in Table 1.
>
> #### Table 1: Per-class results of STONE and AL baselines on nuScene dataset with MinkNet as backbone detector using 5 voxel budgets.
>
> | **Model**    | **Vehicle** | **Person** | **Road** | **Sidewalk** | **Terrain** | **Man-made** | **Vegetation** | **mIoU** |
> |--------------|-------------|------------|----------|--------------|-------------|--------------|----------------|----------|
> | Entropy      | 86.2        | 0.0        | 88.1     | 38.1         | 64.8        | 72.8         | 67.8           | 59.7     |
> | Annotator [1]   | 88.1        | 44.2       | 91.9     | 56.7         | 67.1        | 75.5         | 69.5           | 70.4     |
> | **STONE**    | 85.5        | 34.5       | 81.9     | 41.85        | 65.48       | 72.94        | 70.26          | 64.63    |
>
>
> #### **[W2] Handling of Data Imbalance: Although the paper proposes a two-stage algorithm to handle data imbalance, it does not provide a detailed comparison with other state-of-the-art methods specifically addressing data imbalance in 3D object detection. More in-depth comparative analysis would strengthen the claims regarding its effectiveness in balancing label distribution.**
>
> We adopted the class-balanced grouping [2] into our current pipeline, denoted as STONE-GROUP-BALANCE, which utilizes Focal Loss and dynamically adjusts its weights based on the class distribution within each group during training which won the nuScenes 3D Detection Challenge held in Workshop on Autonomous Driving (WAD, CVPR 2019). We have removed the class balance component of the STONE method accordingly to compare with STONE. Table 2 demonstrates that even with class balancing during training, a highly imbalanced label distribution still results in a performance drop compared to the proposed approach.
>
> #### Table 2: Performance comparisons of STONE and STONE-GROUP-BALANCE using 3D AP(%) scores on the KITTI validation set with SECOND as the backbone architecture.
>
> | **Method**              | **EASY** | **MOD.** | **HARD** |
> |-------------------------|----------|----------|----------|
> | STONE-GROUP-BALANCE     | 74.43    | 63.15    | 57.03    |
> | **STONE**               | **76.86**| **64.04**| **58.75**|
>
>
> #### **[W3] While the paper claims high computational efficiency for STONE, it does not provide specific data on computation time or resource usage. What are the experimental settings and resource usage details?**
> All 3D detection experiments are run on a GPU cluster with 4 NVIDIA RTX A5000 GPUs. In terms of resource usage, STONE consumes only 10 GB of GPU memory compared to 24 GB of memory consumption by KECOR. This is an advantage of 140\% in GPU memory. We will add the details in the final version of the paper.
>
> #### **[Q1]  While the method demonstrates effectiveness in 3D object detection, its generalizability to other domains or tasks is not sufficiently discussed. Is the STONE method applicable to other types of 3D data or different application scenarios?**
>
> Please refer to weakness 1 for additional details.
>
> [1] Xie B, Li S, Guo Q, et al. Annotator: A generic active learning baseline for lidar semantic segmentation[J]. Advances in Neural Information Processing Systems, 2023, 36.
>
> [2]  Zhu B, Jiang Z, Zhou X, et al. Class-balanced grouping and sampling for point cloud 3d object detection[J]. arXiv preprint arXiv:1908.09492, 2019.
>
> [3]  Holger Caesar, Varun Bankiti, Alex H. Lang, Sourabh Vora, Venice Erin Liong, Qiang Xu, Anush Krishnan, Yu Pan, Giancarlo Baldan, and Oscar Beijbom. nuScenes: A multimodal dataset for autonomous driving[J]. CoRR, abs/1903.11027, 2019.

---

> > ### Comment · Reviewer_4dDG · 2024-08-12
> >
> > Thank you for the rebuttal. I am inclined to maintain my original score as it is.

---

> > > ### Author Response · Authors · 2024-08-13
> > >
> > > Dear Reviewer,
> > >
> > > We appreciate your insightful and valuable feedback once again. We will be adding the above experiment results in the camera-ready version.
> > >
> > > Best Regards,
> > >
> > > The Authors

---

### Official Review · Reviewer_jaqa · 2024-07-13

**Soundness:** 3
**Presentation:** 3
**Contribution:** 3
**Rating:** 6
**Confidence:** 3

**Summary:**

This paper proposed an active 3D object framework based on submodular optimization. It focuses on solving data imbalances and covering varying difficulty levels of the point cloud data by using submodular optimization. Extensive experiments show superior results with high computational efficiency.

**Strengths:**

1. The proposed active learning pipeline is intuitive and reasonable.
2. The experiments are thorough. They are conducted across various datasets and show superior results in terms of 3D detection. The ablation study shows the effectiveness of the proposed algorithm.

**Weaknesses:**

1. The paper needs a pipeline figure to illustrate the proposed active 3D object detection. A figure can help better understand the general steps of the proposed approach.
2. Performance gains: Compared with the baseline KECOR, the performance gains seem marginal. L352 mentioned STONE achieves significant GPU memory savings. Why can STONE save significant memory?

**Questions:**

Why only use 1% of labeled bounding boxes? If using more labeled bounding boxes and unlabeled BBOX, is it possible to get better results?

**Limitations:**

The paper has discussed the limitations

---

> ### Author Rebuttal · Authors · 2024-08-06
>
> Thank you for your constructive review and valuable suggestions! Below, we provide a detailed response to your questions and comments.
>
> #### **[W1] The paper needs a pipeline figure.**
>
> Please see the attached PDF for the pipeline figure of the proposed approach.
>
>
> #### **[W2] Compared with the baseline KECOR, the performance gains seem marginal.**
> We further use SECOND [1], a widely used 3D object detector, as the base model.  The results indicate that STONE achieves a **3.4%** higher mAP score at the hard level in 3D detection and **2.43%** higher mAP score at the hard level in BEV detection compared to the state-of-the-art method, KECOR, as shown in Table 1. This demonstrates the performance and generality of the proposed approach.
>
> #### Table 1: 3D mAP(\%) of STONE and AL baselines on KITTI validation set with 1\% queried bounding boxes with one-stage 3D detector backbone SECOND
> | **Methods** | **3D Detection mAP EASY \%** | **3D Detection mAP MOD. \%** | **3D Detection mAP HARD \%** | **BEV Detection mAP EASY \%** | **BEV Detection mAP MOD. \%** | **BEV Detection mAP HARD \%** |
> |----------------|------------------------------|------------------------------|------------------------------|------------------------------|------------------------------|------------------------------|
> | RAND | 69.33±0.62 | 55.48±0.42| 51.53±0.33 | 75.66±1.10 | 63.77±0.86  | 57.71±0.95  |
> | CORESET | 66.86±2.27 | 53.22±1.65 | 48.97±1.42 | 73.08±1.80| 61.03±1.98 | 56.95±1.53 |
> | LLAL           | 69.19±3.43                   | 55.38±3.63                   | 50.85±3.24                   | 76.52±2.24                   | 63.25±3.11                   | 59.07±2.80                   |
> | BADGE          | 69.92±2.90                   | 55.60±2.72                   | 51.23±2.58                   | 76.07±2.70                   | 63.39±2.52                   | 59.47±2.49                   |               |
> | CRB            | 72.33±0.35                   | 58.06±0.30                   | 53.09±0.31                   | 78.84±0.27                   | 65.82±0.07                   | 61.25±0.22                   |
> | KECOR          | 74.05±0.16                   | 60.68±0.13                   | 55.34±0.23                   | 80.00±0.12                   | 68.20±0.35                   | 63.26±0.25                   |
> | **STONE**      | **76.86±0.88**               | **64.04±0.27**               | **58.75±0.58**               | **82.14±0.90**               | **70.82±0.14**               | **65.68±0.42**               |
>
> #### **[W3]  L352 mentioned STONE achieves significant GPU memory savings. Why can STONE save significant memory**
> Given the point cloud $\mathcal{P}_i$, a typical 3D object detector first leverages an encoder $\mathbf{g}(\cdot; \theta_g)$ for extracting the latent feature $\mathbf{m}_i$, then a detector head $\mathbf{h}( \cdot; \theta_h)$ is used to generate detection results $\hat{\mathcal{B}_i}$. KECOR needs to compute gradients of the output of the ROI head's fully connected shared layer, i.e., $\mathbf{m}_i$, with respect to the encoder parameters. The gradients are a matrix of high dimensions.  In contrast, STONE only needs to compute the gradient of the output of the ROI head's classification loss layer and regression loss layer, i.e., $\hat{\mathcal{B}_i}$, with respect to the encoder parameters. Therefore, STONE can save significant memory compared with KECOR.
>
> #### **[Q1] Why only use 1% of labeled bounding boxes? If using more labeled bounding boxes and unlabeled BBOX, is it possible to get better results?**
> To ensure a fair comparison with the previous state-of-the-art methods, we leveraged 1\% of the labeled bounding boxes. Referring to Tables 2 and 3, as more labeled bounding boxes are added to the training, the results get better. It is worth noting that the KECOR method marginally surpasses STONE when using 2\% and 3\% of the bounding box. This is because the STONE method, when utilizing the PV-RCNN backbone, tends to select scenes with more objects. As a result, STONE reaches the bounding box limit very early in the active learning stage. The slightly better results achieved by KECOR are due to it querying more scenes and being trained over more epochs. In active 3D object detection, the goal is to label as few bounding boxes as possible to achieve good performance. Therefore, it is critical to maintain high performance with a small number of labelled bounding boxes, as demonstrated by STONE. When using SECOND [1] as the backbone detector, STONE is consistently better than other state-of-the-art AL baselines (Table 3) even when using more bounding boxes.
>
> #### Table 2: Performance comparisons of STONE and AL baselines using 3D AP(%) scores on the KITTI validation set (HARD level) with PV-RCNN as the backbone architecture.
>
> | Method | 3D AP \% using 1% (bounding box) |  2\% |  3\% |
> |--------|---------------------------|--------------|--------------|
> | CRB    | 62.81                     | 65.43        | 69.93        |
> | KECOR  | 63.42                     | **67.25**    | **71.70**    |
> | **STONE** | **64.05**              | 66.83        | 70.86        |
>
> #### Table 3: Performance comparisons of STONE and AL baselines using 3D AP(%) scores on the KITTI validation set (HARD level) with SECOND [1] as the backbone architecture.
>
> | Method | 3D AP \% using 1% (bounding box)    | 2\% |  3 \% |
> |--------|---------------------------|--------------|--------------|
> | CRB    | 53.09                     | 55.67        | 57.01        |
> | KECOR  | 55.34                     | 57.56        | 58.92        |
> | **STONE** | **58.75**              | **60.33**    | **61.89**    |
>
> [1] Yan Y, Mao Y, Li B. Second: Sparsely embedded convolutional detection[J]. Sensors, 2018, 18(10): 3337.

---

> > ### Comment · Reviewer_jaqa · 2024-08-08
> >
> > I appreciate the author's response. It has addressed my concerns.

---

> > > ### Author Response · Authors · 2024-08-13
> > >
> > > Dear Reviewer,
> > >
> > > We appreciate your insightful and valuable feedback once again. We will be adding the above experiment results in the camera-ready version.
> > >
> > > Best Regards,
> > >
> > > The Authors

---

### Author Rebuttal · Authors · 2024-08-07

Figures for Rebuttal

---

### Decision · Program_Chairs · 2024-09-25

**Decision:**

Accept (poster)

**Comment:**

This paper introduces a method for active object detection in 3D data based on submodular optimization, reducing labelling costs. It received favorable reviews from five experts, who appreciated the idea and novelty, SOTA performance on several datasets, breadth, presentation and writing.

Some weaknesses were raised, generality of submodular functions, significance of gains, missing information, requests for some additional experiments, a single baseline detector, minor doubts on experiments, some missing references.

The rebuttal provided by the authors could address several concern and provided additional experiments, including on semantic segmentation, which was satisfactory for all reviewers.

A consensus on publication was quickly reached, the AC concurs.